# Attitudes of Patients with Chronic Diseases toward Management eHealth Applications Systems in Post-COVID-19 Times

**DOI:** 10.3390/ijerph19074289

**Published:** 2022-04-03

**Authors:** Abdullah H. ALsharif

**Affiliations:** Department of Management Information Systems, College of Business Administration-Yanbu, Taibah University, Medina 42353, Saudi Arabia; alsharifa@taibahu.edu.sa

**Keywords:** post-Covid-19, chronic diseases, eHealth, eHealth applications

## Abstract

Introduction: There has been an increase in the adoption of eHealth technologies and applications by health consumers globally because of the restrictions imposed due to the COVID-19 pandemic in the last two years. The sudden change in the users’ attitudes toward eHealth adoption needs to be critically evaluated and understood, as it can be the stepping stone toward rapid digitalization of healthcare operations in Saudi Arabia as a part of Vision 2030. Purpose: The purpose of this study was to evaluate the attitudes of the patients with chronic diseases toward eHealth applications in post-COVID times. Methods: A cross-sectional study design was adopted using the online questionnaire as a data collection instrument. All the health consumers using eHealth services aged above 18 years and living in Saudi Arabia were included in the survey. The survey was conducted for 3 weeks, resulting in a final sample of 234 participants. Results: Overall, 73.8% of the participants stated that they adopted eHealth only out of necessity, while 37.3% stated that they adopted it because no other services were available. Only 10.3% stated that they adopted eHealth out of interest. In relation to the future use of eHealth, 51.5% of the participants stated that they would definitely not use eHealth applications, and 33.6% stated that they would probably them once the pandemic ends. Only 4.4% of the participants stated that they would very much probably, and 10.5% stated they would probably not use eHealth applications once the pandemic ends. a significant difference in opinions in relation to the future adoption of eHealth applications was observed among the male and female participants, and also between the age groups of younger (age <35 years) and older (age ≥35 years) participants. Conclusions: For the change in attitudes (increased adoption of eHealth) to be sustained, policymakers need to develop relevant strategies promoting the use of eHealth in Saudi Arabia.

## 1. Introduction

Management information systems (MIS) play a huge role in the heart of E-health systems particularly during pandemics. Where since its discovery in December 2019, the novel coronavirus (SARS-CoV-2) has mutated into various variants, which has led to different waves of surges of COVID-19 in different nations. As of 15 December 2021, 270.7 million confirmed COVID-19 cases were identified globally, resulting in 5.3 million deaths. Currently, more than 8 billion vaccine doses have been administered globally [1]. While some counties in Europe are already experiencing the fourth wave of infections due to the new variant Omicron, a few countries such as India, Brazil are expecting a third wave of infections [2]. Saudi Arabia is one of the largest countries in the Middle East, which has experienced two waves of infections from April 2020 to September 2020, and from March 2021 to August 2021 [3]. Thus far, there have been 550,369 confirmed cases, and 8856 deaths related to COVID-19 have been recorded in Saudi Arabia since December 2019 [3]. Saudi Arabia has experienced a series of lockdowns and curfews as a part of preventive measures in order to contain the spread of the novel coronavirus since its identification in January 2020 in the country [4].

As a part of vision 2030, Saudi Arabia has been implementing digitalization in healthcare [5], which has become a boon during the COVID-19 pandemic. Though the digitalization process has ramped up, studies [6,7] have identified issues such as lack of preparedness and awareness of eHealth among the citizens, which has affected the implementation of eHealth solutions in the country. A recent study [8] has highlighted that lack of facilitating conditions in relation to eHealth has affected the trust among healthcare consumers. Furthermore, there were issues identified in relation to the formulation of eHealth policies and implementing them, as disparities were identified among the various population groups in terms of accessibility, utilization, and perceptions of digital health technologies [9].

Various studies conducted during the COVID-19 pandemic have shown similar results with respect to the intention to use eHealth, whereas contrasting results were identified before the pandemic in studies conducted in different countries. For instance, factors such as social reference, advertisement, attitudes toward the system, access to mobile phones, and perceived system effectiveness were identified to be the most influential factors affecting the intention to use the eHealth system among rural citizens in Bangladesh [10]. Similarly, trust, privacy, and social influence were the main factors identified that influence the intention to use mHealth technologies in Indonesia [11]. Similarly, social influence, facilitating conditions, hedonic motivation, price value, and habit were identified to be the most influential actors in adopting eHealth in France [12]. Social influence and facilitating conditions were identified as having high influences on the users in most of the studies [10,11,12] during the pandemic, implying higher social impact and a lesser effect of the self-decision-making approach. As further support for this statement, performance expectancy had no impact on the intention to use eHealth in a study conducted in Turkey [13]. Furthermore, all stakeholders, including physicians [14,15], patients, and especially older adults [16,17,18], reflected an increased use of eHealth applications in different countries. 

Similarly, focusing on the current eHealth framework adopted in Saudi Arabia, the need for additional components was recognized, focusing mainly on effective management of information, extensive stakeholder engagement, promoting awareness, especially in self-management and self-collaboration in managing chronic illnesses, and increasing accessibility and reachability [19]. Studies focusing on the acceptance of eHealth applications have identified several interesting aspects. Perceived usefulness and ease of use were the two major factors influencing the attitudes of the people in Saudi Arabia toward eHealth usage. Furthermore, subjective norms were identified to be significantly influencing the behavioral intention to use eHealth applications [20]. Furthermore, lack of sufficient skills and competencies and lack of willingness to use eHealth technologies among healthcare practitioners were also identified to be important factors affecting the use of eHealth applications by patients [21]. Among the users who adopted eHealth applications, issues such as cloud storage, platforms, quality of service (QoS), security, and data acquisition affected their continuous usage of eHealth applications [22,23]. Furthermore, demographic factors such as age, gender, residence, income, education, and culture were identified to be the major factors that would hinder the adoption of the eHealth system [24,25].

However, a significant rise in the number of eHealth users was observed in the country in the past few years. The number of eHealth users increased from 7.1 million in 2017 to 9.7 million in 2021. Online doctor consultations increased from 0.35 million in 2017 to 1.14 million in 2021, whereas eHealth apps users increased from 0.67 million in 2017 to 0.91 million in 2021. Significant growth in the adoption of eHealth devices users was observed, which rose from 0.53 million in 2017 to 1.14 million in 2021. Furthermore, online revenue generation from eHealth devices and online doctors’ consultation per user was identified to be USD 56.2 and USD 87.02, respectively, in 2017; this increased to USD 72.29 and USD 105.69 in 2021, respectively, and is projected to reach USD 89.48 and USD 124.37 by 2025, respectively [26]. Apart from the government efforts to increase the digitization of healthcare services, COVID-19 can be one of the major reasons for the rapid increase in eHealth adoption in the past few years, though previous studies have identified a lack of awareness and several other issues affecting the adoption of eHealth in Saudi Arabia. Currently, the user penetration rate in eHealth in Saudi Arabia stands at 36.51, which is very low, compared with the UK (53.05), the US (47.89), but very high compared with other countries in Asia such as India (12.18) and Egypt (14.05) in the Middle East [26], indicating a fairly better position in terms of eHealth adoption globally. Furthermore, a recent study [27] has indicated that there is a continual growth in both publications and eHealth awareness and its significance in Saudi Arabia. Accordingly, supporting the claims, the government has developed and launched various mHealth applications such as Mawid, Sehha, Ashanak, Mawared, etc. during the COVID-19 pandemic in order to provide remote support to healthcare consumers [28,29]. However, it was identified that necessity—but not interest in eHealth—as well as fear, stress, depression, and anxiety, were identified to be more influential in increasing adoption of eHealth during the pandemic rather than factors such as ease of use, usefulness, and enjoyment factors [30], which were identified to be more influential in eHealth adoption in previous studies.

Therefore, the underlying reason behind the behavioral and attitudinal change in patients in relation to the eHealth adoption during the pandemic is unclear—namely, whether it is a change led by motivation and awareness about eHealth benefits, or whether it is a change led by fear, necessity, and anxiety during the COVID-19 pandemic. As the Saudi government is aiming for complete digitization of healthcare services as a part of Vision 2030, it is very important that changes in attitudes of healthcare consumers be studied at regular intervals, in order to develop policies and strategies, and effectively implement them. Therefore, it is very important to identify the attitudes of the healthcare consumers in Saudi Arabia in current times, where the COVID-19 restrictions have been greatly relaxed, in order to determine the change in attitudes and behavior of healthcare consumers in relation to eHealth adoption. Considering these factors, the purpose of this study is to evaluate the attitudes of the patients with chronic illnesses toward eHealth applications in the post-COVID or ongoing COVID-19 situation, where the restrictions such as lockdowns and curfews are greatly relaxed in the country.

## 2. Methods

This exploratory study used a cross-sectional survey design for data collection and analysis. Considering the ongoing COVID-19 protocols to maintain social distancing, an online platform was used for conducting the survey.

### 2.1. Questionnaire Design

The questionnaire is divided into two sections as shown in Appendix A. The first section focuses on the demographic information of the participants and contains six questions. The second section contains 18 questions focusing on the main research-related questions. The first question relates to the participant’s current health condition, and the second question is related to the future actions that participants would take to manage their health condition. The third question focuses on the confidence levels of the participants in self-management of their health condition. The fourth question analyzes the sources of health information of participants. The fifth question focuses on the participants’ comfort in using health information. The sixth question focuses on participants’ online search behavior for health information, and the seventh question focuses on the participants’ interest in using eHealth. The eighth question focuses on the preferences of participants in tracking their health. The ninth question focuses on the participants’ usage of eHealth applications for various services. Questions 10–11 focus on participants’ preferences for eHealth applications. Question 12 focuses on the participants’ perceptions about the issue of privacy and security. Question 13 focuses on the frequency of usage, and question 14 focuses on the services accessed using eHealth applications. Question 15 focuses on the effectiveness of using mobile phones for accessing eHealth applications. Question 16 focuses on features of eHealth applications. Questions 17–18 focus on the impact of the pandemic on participants’ current and future eHealth usage. The survey was designed online using the Google Surveys platform, for which an online link was generated.

The survey questionnaire was initially developed by authors in English, which was later translated into Arabic using two professional translators. A pilot study was conducted with 12 randomly selected participants in order to check the reliability of the questionnaire. The pilot study results were analyzed, and the Cronbach alpha for all items was identified to be greater than 0.70, indicating good reliability [31]. 

### 2.2. Sampling and Participants

As the survey is targeted at eHealth users who use the relevant services, only adults aged above 18 years are included in the study. Furthermore, all the people living in Saudi Arabia and accessing eHealth services were included in the study. Convenience sampling [32,33] was used in selecting the participants, as it was important for the researchers to consider the participants who can be easily accessed due to COVID-19 conditions. The survey link was forwarded to the participants on community pages of eHealth users and also on health and fitness influencer pages (@tabibgroup; @nutters.sa, etc.) on various social media platforms including Facebook, Twitter, and Instagram. In addition, WhatsApp was used to personally forward the survey link to authors’ colleagues, families, and friends. The survey was conducted for three weeks from 12 October 2021 to 2 November 2021. At the end of the survey, a total of 398 responses were received, out of which, 138 participants submitted incomplete responses. Therefore, a final sample of 234 participants and their responses were considered for data analysis.

### 2.3. Data Analysis

As the data included both descriptive and statistical information, the data were analyzed descriptively, in addition to using statistical methods such as *t*-tests in order to identify the difference in opinions between participant groups. IBM SPSS version 25 (IBM, Portsmouth, UK) was used for analyzing the data.

### 2.4. Ethical Considerations

All of the participants were fully informed about the purpose of the study, and informed consent was taken from them before participating in the survey using the check box provided for “agree to participate” in the survey. Furthermore, participation in the survey was voluntary, and the participants could leave the survey at any time if they did not want to continue. Ethical approval for the study was received from Taibah University, Saudi Arabia.

## 3. Results

The total number of participant responses considered for data analysis was 234. The demographic information of the participants is presented in Table 1. Male participants were identified to be 53.4% of the total participants, while female participants included 46.6%, reflecting an appropriate representation of both genders in the survey. The education levels shown in Table 1 are representative of the total sample population. The majority of the participants graduated with a bachelor’s degree (55.9%), followed by 24.8% of participants with high school/diploma qualification, while 11.5% graduated with a master’s degree, and 7.7% have PhD. Similarly, the majority of the participants were aged 18–24 years, followed by 19.2% aged 35–44 years, 14.9% aged 25–34 years, 6.8% aged 45–54 years, and 3.8% above or equal to 55 years. 

Regarding the usage of eHealth applications, 76.5% of the total participants are currently using them, whereas 23.5% had used them before but are no longer using them. The overall health condition of the participants was identified to be very good, as the majority of the participants stated their overall health condition as excellent (32.5%) and very good (51.7%). Regarding the intention to use these applications in the next six months, about 35.9% of participants stated that they would continue to use their existing eHealth applications, while 24.4% of participants stated that they would install a new eHealth application. It is interesting to note that 39.7% of participants stated that they would take no action, i.e., they would not use any eHealth application. 

When asked how confident they were to manage their condition by themselves, the majority of the participants (59%) stated that they were somewhat confident, while 29.9% stated that they were very confident. Furthermore, 10.3% of participants stated that they were not too confident. 

As evident from Figure 1, the majority of the participants acquired a considerable amount of health information from the internet, family, and friends, eHealth applications, and social media. The internet (71.5%) was used more frequently than eHealth applications (59.7%) for finding health-related information by the participants. Furthermore, in relation to the participants’ attitudes toward searching for information, the following percentages of participants disagreed with the corresponding statements: “I know exactly what it is that I want to learn about my health” (48.4%); “I can figure out how and where to get health information I need” (39.2%); “I am satisfied with the way I currently learn about health” (51.8%); “I feel that I am in control over how and what I learn about health” (49.6%).

As observed in Table 2, the majority of the participants used online platforms for searching for information about a disease. However, other activities such as accessing test/diagnostic results, consulting doctors, renewing prescriptions, and posting about health status were identified to be the other major online activities of the participants.

As observed in Figure 2, only 37.6% of the participants were very interested in using eHealth applications, while the majority (44.4%) were somewhat interested, and 15% were not too interested in using eHealth applications. In relation to the tracking feature in eHealth applications, the majority of the participants were interested to track diet and calories intake (62%) and exercise (50.4%).

As observed in Table 3, the majority of the participants either strongly accepted or accepted the use of the above-listed activities with eHealth applications, indicating a strong interest in eHealth. In relation to the influencing factors of privacy and security, only 15% of the participants stated that they are not worried, while 19.3% stated that they are very worried, 33.9% stated that they are somewhat worried, and 31.8% stated that they are not too worried. 

As observed in Figure 3, the majority of the participants used eHealth applications once a month (38.9%), while 27% used them at least once a week, and 25.7% used them at least once a day. Only 8.4% of the participants used eHealth applications many times a day. 

In relation to the service sought by the participants from eHealth applications, 68.4% preferred primary healthcare service (diagnosis, health advice, and treatment), 36% preferred specialized healthcare service (remote monitoring using wearables, treatment), and 56.6% preferred health education/information. More than 90% of the participants preferred mobile/smartphones for using eHealth applications. About 48.2% of participants disagreed with the statement that eHealth applications offer greater security, while only 27.3% agreed with it. Similarly, 53.4% disagreed with the statement that it is easy to share information on eHealth applications (Figure 4), while only 24.7% agreed with it. It is interesting to note that, 73.8% of the participants stated that they adopted eHealth only out of necessity, while 37.3% stated that they adopted it because no other services were available. Only 10.3% stated that they adopted eHealth out of interest. Further analyzing the results in terms of gender and age groups, interesting findings were revealed. As shown in Table 4, no significant difference in opinions was observed among male and female participants, and also between the age groups of younger (age <35 years) and older (age ≥35 years) participants. 

In relation to the future use of eHealth, 51.5% of the participants stated that they would definitely not, and 33.6% stated that they would probably use eHealth applications once the pandemic ends. Only 4.4% of the participants stated that they would very much probably, and 10.5% stated they would probably not use the eHealth applications once the pandemic ends.

Further analyzing the results in terms of gender and age groups, interesting findings were revealed. As shown in Table 5, a significant difference in opinions in relation to the future adoption of eHealth applications was observed among male and female participants, and also between the age groups of younger (age <35 years) and older (age ≥35 years) participants. Male participants and younger participants were more inclined toward not using the eHealth application, compared with female and older participants, respectively.

## 4. Discussion

As evident from the results, the majority of the participants were using eHealth applications at the time or in the past. However, their primary source of collecting health-related information was mainly the internet, as well as family and friends, while the eHealth application was used only by 59.7% for accessing health-related information. These findings were similar to a study [34], which identified the internet (44.5%), and family and friends (45.7%) as the major sources of health information for participants in Saudi Arabia. Other studies [11,15,16] identified the influence of culture and social norms on the behavioral attitudes of the health consumers toward eHealth, and in accordance with their findings, in this study, it was also observed that the internet, and family and friends, were considered to be more reliable sources of information than the information available on eHealth applications. Accordingly, analyzing the online behavior, the majority of the participants reflected activities such as information searching about the disease and consulting doctors as their major activities. Accordingly, another study [34] identified doctors are considered to be the most trusted and reliable sources of information, followed by pharmacists in Saudi Arabia. Analyzing the participants’ interest in using the eHealth applications, the majority of them were somewhat or not at all interested. These may be attributed to the lack of awareness and preparedness among healthcare consumers [6,7], and also the lack of sufficient skills, training, and support among healthcare professionals [21]. Further support for this result was that the majority of the participants stated that they use eHealth applications at least once a month. Therefore, it can be discerned that the eHealth applications for daily health tracking may not be used by the majority of the participants, while they may be used for other purposes such as booking appointments with doctors or ordering medicine online, as observed in [26]. This can be further supported by the finding that over 60% of the participants used eHealth applications for primary healthcare services. 

Previous studies [24,35,36,37] have identified privacy, security, lack of support and training, etc. to be the major issues affecting the adoption of eHealth applications in Saudi Arabia. Focusing on the issues of concerns in relation to eHealth applications, privacy and security continued to be the major concerns among healthcare concerns, as observed in this study. Focusing on the eHealth adoption in recent times, the majority of the participants stated that they adopted either out of necessity or because there was no other alternative, clearly reflecting the impact of the COVID-19 pandemic on the surge in eHealth adoption in Saudi Arabia. Accordingly, studies in different countries [10,11,12,13,14,15,16,17,18,19] reflected an increased adoption of eHealth technologies, especially mHealth applications, for consultation, appointment booking, medicine orders, remote monitoring of chronic conditions such as diabetes, etc. However, research identifying the intention to use eHealth technologies in post-COVID-19 times in various countries is lacking. In the context of Saudi Arabia, as analyzed in this study, only 10.3% of the participants adopted eHealth out of interest. Further analysis of the differences in opinions revealed no significant differences among both genders, as well as between younger and older populations. In terms of participation intention to use eHealth in the future, it is interesting to note that nearly half of the participants stated they would not use the eHealth application in the future. Supporting these findings, out of 76.5% who currently use the eHealth application, 51.9% stated they would do nothing in the next six months, i.e., they would not use the eHealth application. Previous studies in Saudi Arabia [24,38] have found various influencing factors affecting the adoption of eHealth technologies, which include bureaucracy, absence of clear policies, lack of information security, limitations in the operation and maintenance of ICTs, socio-cultural issues, etc. Socio-cultural issues such as gender, social norms, and religion were identified as the major factors influencing eHealth adoption. The results in this study reflect that the socio-cultural influence still exists, as differences in opinions were observed among the genders and different age groups.

The findings in this study thus indicated a change in the attitudes of healthcare consumers toward eHealth, mainly influenced by the pandemic and slightly influenced by self-interest in adoption. These findings can have both practical and theoretical implications. Firstly, they can aid policymakers and decision makers in understanding the impact of the pandemic on the change in attitudes of health consumers toward eHealth and accordingly design promotional strategies to increase the adoption of eHealth as a part of the Vision 2030 digitization plan. Secondly, the findings contribute to the literature by interlinking pandemic, health consumers’ attitudes, and eHealth adoption, in continuance to previous studies. The main focus of this study was to investigate if the rapid increase in the adoption of eHealth in Saudi Arabia was due to self-interest or out of necessity. It is very important to identify this aspect, as it is going to affect the use of eHealth in the future in Saudi Arabia. With Vision 2030, Saudi Arabia aims to completely transform its healthcare infrastructure, and therefore, it is very important to assess if its population is ready and prepared for such rapid digitized healthcare services. The findings in this study offer insights into the attitudes of eHealth users—namely, that the change observed during the pandemic was due to self-interest but out of necessity. This can help them in developing awareness programs motivating the users to continue using eHealth technologies. 

This study also has a few limitations. The sample identified in this study was comparatively low; therefore, the generalization of findings in this study has to be carried out with care. Although the response rate was sufficient, a large number of participants submitted incomplete responses due to which the final sample for data analysis was comparatively low. As a result, the number of the younger population represented in the sample was high. Another limitation of this study was that most of the results were descriptively analyzed; therefore, future studies can focus on quantitative analysis using different statistical techniques. Furthermore, usually, the older population is associated with usability issues in eHealth applications [39,40]; therefore, in this context, future studies may focus on the usability of eHealth applications among older adults in post-COVID19 times.

## 5. Conclusions

The main purpose of conducting this study was to analyze the change in users’ attitudes toward eHealth. It is important to understand this aspect because, if the change in the users’ attitudes is influenced by the COVID pandemic, and the users’ increasing adoption of eHealth is motivated by necessity, the change may not sustain after the pandemic ends. The findings of this study clearly demonstrated that the change in users’ attitudes toward eHealth adoption is greatly influenced by necessary conditions due to the pandemic. Therefore, it is important that the government and policymakers adopt strategies to sustain this change, as it has been already initiated by increasing the adoption of eHealth solutions. 

## Figures and Tables

**Figure 1 ijerph-19-04289-f001:**
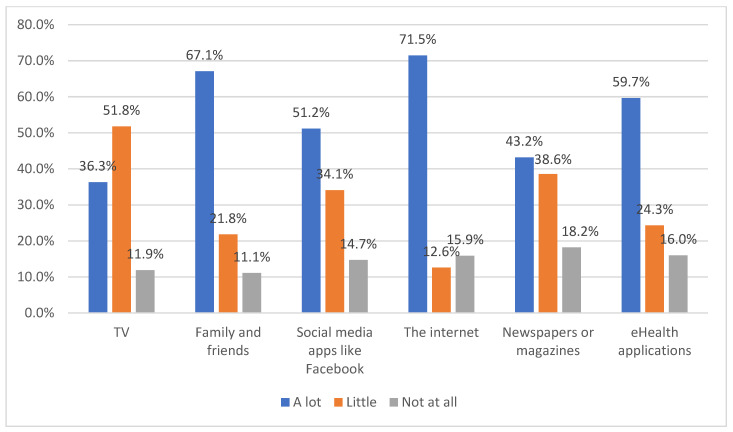
Sources of health information.

**Figure 2 ijerph-19-04289-f002:**
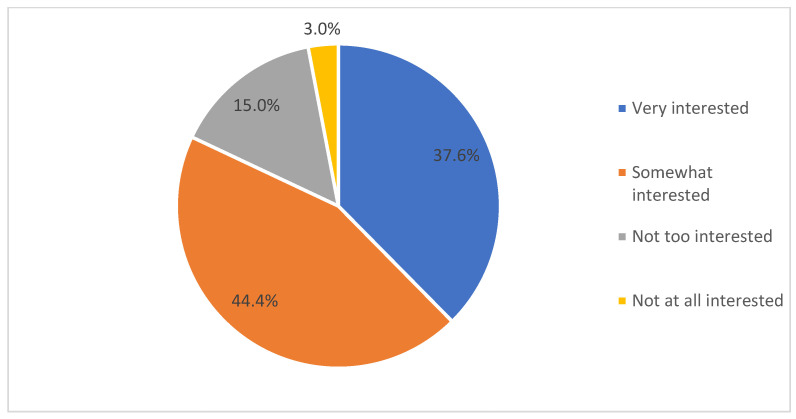
Participants’ levels of interest in using eHealth applications.

**Figure 3 ijerph-19-04289-f003:**
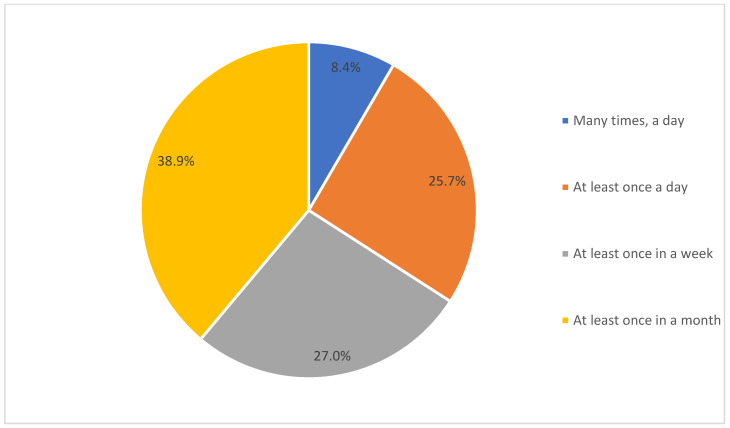
Participants’ frequency of using eHealth applications.

**Figure 4 ijerph-19-04289-f004:**
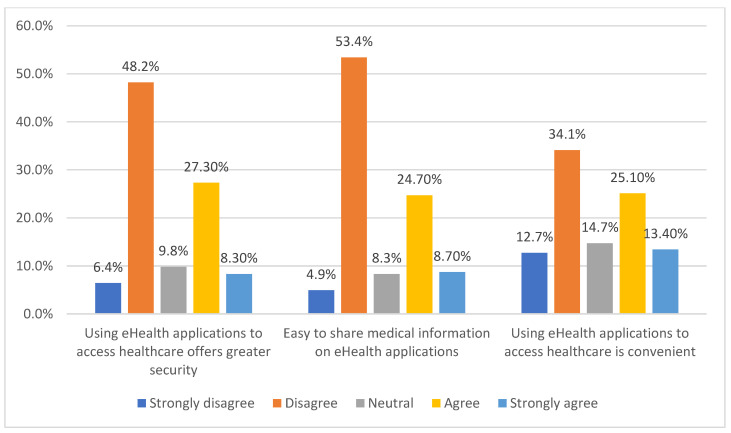
Uses of eHealth applications.

**Table 1 ijerph-19-04289-t001:** Participant demographics.

Demographic Characteristics	N	Relative Frequency
**Gender**		
Male	125	53.4%
Female	109	46.6%
**Education**		
High School/Diploma	58	24.8%
Bachelor’s degree	131	55.9%
Master’s degree	27	11.5%
Doctorate	18	7.7%
**Age (years)**		
18–24	129	55.1%
25–34	35	14.9%
35–44	45	19.2%
45–54	16	6.8%
≥55	9	3.8%

**Table 2 ijerph-19-04289-t002:** Online behavior of the participants.

Online Activities	Frequency
Searched online for information about a disease or medical problem	64.2%
Searched online for information about a doctor	28.7%
Typed information on an application about what you eat, how much you exercise, or your weight	38.7%
Typed information on an application about a chronic illness you have	37.2%
Renewed a prescription online	57.6%
Consulted your doctor	63.5%
Used a personal health record	29.4%
Looked at a test result online	54.1%
Used a device that measures health information (like blood pressure; blood glucose levels) that connects to your mobile/website application	56.3%
Posted anything online about your health or health care	49.9%
Joined an online group that is for a health issue that you or your family member has	34.6%
Booked appointment with doctors	53.8%

**Table 3 ijerph-19-04289-t003:** Participants’ levels of acceptance for using the following activities on eHealth applications.

Activities	Mean
Booking of appointments with physicians	4.34
Access to laboratory test results	4.25
Provision of educational resources	4.21
Electronic renewal of prescriptions	4.21
Personal repository of medical documentation	4.22
Regular reporting of disease status to physician	4.23
Contact with healthcare provider in case of disease exacerbation	4.26
Online contact with healthcare professional (nurse or physician) on an as-needed basis	4.19
Referral to physician	4.30

Ratings: 5—strongly accept; 4—accept; 3—neutral; 2—reject; 1—strongly reject.

**Table 4 ijerph-19-04289-t004:** Difference in opinions among participants groups in relation to eHealth adoption reasons.

	N	Mean	Standard Deviation	*df*	*T-Value*	*p-Value*
	By gender					
Male	125	2.4	3.14	232	0.7782	0.4372
Female	109	2.65	1.26			
	By age					
<35 years	164	2.55	1.68	232	0.1501	0.8808
≥35 years	70	2.5	3.41			

**Table 5 ijerph-19-04289-t005:** Difference in opinions among participants groups in relation to eHealth adoption in future.

	N	Mean	Standard Deviation	*df*	*T-Value*	*p-Value*
	By gender					
Male	125	2.04	0.43	232	2.4458	**0.0152 (*p < 0.05*) ***
Female	109	1.84	0.79			
	By age					
<35 years	164	1.99	0.38	232	2.0230	**0.0442 (*p < 0.05*) ***
≥35 years	70	1.85	0.67			

***** Bold: Significant difference.

## Data Availability

The data presented in this study are available on request from the corresponding author. The data are not publicly available due to requirement for participants’ consent.

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
