# Peer review of "Attitudes of Patients with Chronic Diseases toward Management eHealth Applications Systems in Post-COVID-19 Times"

_ijerph, 2022, doi:10.3390/ijerph19074289_

Round 1

Reviewer 1 Report

Most of the earlier comments have been dealt with in an adequate way. 

Two issues remain. There is a need to stress that this is an exploratory study and (second) using an convenience sample. This should be done both in the methods section als well as in the discussion. 

Author Response

Reviewer 1

Comment/s

Explanation

Need to stress, this is an exploratory study

“This exploratory study used a cross-sectional survey design for data collection and analysis.”

-          Added in Methods section

Stress using convenience sample

“Convenience sampling [40] was used in selecting the participants, as it was important for the researchers to consider the participants who can be easily accessed due to Covid-19 conditions”

-          Added in sampling and participants section

Reviewer 2 Report

While I still have some reservations about the descriptive nature of the study with no propositions to test in the future, you have addressed these concerns in the revised paper.  I still think the focus of e-health is more important in addressing the chronic care needs of older population as they have the most barriers and contribute most to the costs. Wish you focus much of your future work, with this insight, on how to make e-health viable to address their needs. 

Author Response

Reviewer 2

No propositions to test in future

Added in discussion

Future propositions with respect to older population

“Furthermore, usually older population are associated with usability issues in eHealth applications [41,42], therefore, in this context, future studies may focus on the usability of eHealth applications among older adults in post-Covid19 times.”

-          Added in discussion